# Nontuberculous Mycobacteria Prevalence in Bats’ Guano from Caves and Attics of Buildings Studied by Culture and qPCR Examinations

**DOI:** 10.3390/microorganisms9112236

**Published:** 2021-10-27

**Authors:** Ivo Pavlik, Vit Ulmann, Helena Modra, Milan Gersl, Barbora Rantova, Jan Zukal, Katerina Zukalova, Ondrej Konecny, Vlastislav Kana, Pavel Kubalek, Vladimir Babak, Ross Tim Weston

**Affiliations:** 1Faculty of Regional Development and International Studies, Mendel University in Brno, Tr. Generala Piky 7, 613 00 Brno, Czech Republic; helena.modra@mendelu.cz (H.M.); ondrej.konecny@mendelu.cz (O.K.); 2Public Health Institute Ostrava, Partyzanske Nam. 7, 702 00 Ostrava, Czech Republic; vit.ulmann@zuova.cz; 3Faculty of AgriSciences, Mendel University in Brno, Zemedelska 1/1665, 613 00 Brno, Czech Republic; milan.gersl@mendelu.cz (M.G.); barbora.rantova@mendelu.cz (B.R.); 4Institute of Vertebrate Biology of the Czech Academy of Sciences, v.v.i., Kvetna 8, 603 65 Brno, Czech Republic; zukal@ivb.cz; 5Faculty of Veterinary Hygiene and Ecology, University of Veterinary Sciences Brno, Palackeho Tr. 1946/1, 612 42 Brno, Czech Republic; H20363@vfu.cz; 6Museum Blanenska, Zamek 1/1, 678 01 Blansko, Czech Republic; prirodovedec@muzeum-blanenska.cz; 7Central Bohemian Archaeological Heritage Institute, Nad Olsinami 448/3, 100 00 Prague, Czech Republic; pavel.kubalek@seznam.cz; 8Veterinary Research Institute, v.v.i., Hudcova 70, 621 00 Brno, Czech Republic; babak@vri.cz; 9Department of Biochemistry and Genetics, La Trobe Institute for Molecular Science, La Trobe University, Bundoora, Melbourne, VIC 3086, Australia; R.Weston@latrobe.edu.au

**Keywords:** non-tuberculous mycobacteria (NTM), mycobacteria other than tuberculosis (MOTT), saprophytic environmental mycobacteria, risk groups of microorganisms, microchiroptera

## Abstract

A total of 281 guano samples were collected from caves (N = 181) in eight European countries (Bulgaria, Czech Republic, France, Hungary, Italy, Romania, Slovakia and Slovenia) and attics in the Czech R. (N = 100). The correlation of detection of mycobacteria between Ziehl–Neelsen (ZN) microscopy and culture examination and qPCR was strong. ZN microscopy was positive in guano from caves (58.6%) more than double than positivity in guano from attics (21.0%; *p* < 0.01). From 89 mycobacterial isolates (73 isolates from cave guano and 16 isolates from attics’ guano), 68 (76.4%) isolates of 19 sp., ssp. and complex were identified as members of three Groups (*M. fortuitum*, *M.*
*chelonae*, and *M. mucogenicum*) and four complexes (*M. avium*, *M. terrae*, *M.*
*vaccae*, and *M.*
*smegmatis*). A total of 20 isolates (22.5%) belonged to risk group 1 (environmental saprophytes), 48 isolates (53.9%) belonged to risk group 2 (potential pathogens), and none of the isolates belonged to risk group 3 (obligatory pathogens). When comparing bat guano collected from caves and attics, differences (*p* < 0.01; Mann–Whitney test) were observed for the electrical conductivity, total carbon, total organic, and total inorganic carbon. No difference (*p* > 0.05; Mann–Whitney test) was found for pH and oxidation-reduction potential parameters.

## 1. Introduction

Bats are well known for being reservoirs for many different pathogenic agents [1,2,3]. Most of the recent interest in this field has been focused on viruses [4,5,6,7], however, bats can also host a wide range of fungi [8,9] and protozoa [10,11]. While knowledge of presence of different pathogens in bats continues to grow, very little is known about the pathogenicity of bacteria in bats [2,12]. Previous studies have shown bats carrying *Anaplasma phagocytophilum* [13], *Bartonella* spp. [14,15,16,17,18], *Coxiealla* spp. [19], *Enterococcus faecalis* [20], *Leptospira* spp. [15,21], mycoplasmas [22,23], *Pasteurella multocida* [24], Rickettsia [18], *Staphylococcus nepalensis* [25], or *Streptomyces* sp. [26]. Antibiotic-resistant bacteria from bat faecal microbiome was studied in *Myotis myotis* and *Rhinolophus hipposideros* in Slovakia [27].

However, there is a clear lack of knowledge on the presence of species of the family Mycobacteriaceae in bats [2,28]. The first reports describing *Mycobacterium bovis* isolation from an Indian Flying fox (*Pteropus giganteus*) living in captivity in zoological gardens in the UK were published between 1925 and 1931 [29,30,31]. The anamnestic data of all these three cases revealed that the animals were occasionally fed condensed milk, which could naturally be contaminated with *M. bovis* from infected cows. Subsequently, non-photochromogenic nontuberculous mycobacteria (NTM) belonging to III Runyon Group [32] were isolated from the livers of 10 Brazilian free-tailed bats (*Tadarida*
*brasiliensis*) in Eagle Creek in the USA in 1969 [8].

Bats have also been used as an experimental animal model for mycobacterial infections. In 1963, successful infection in the footpad of Eastern Pipistrelle (*Pipistrellus subflavus*) and Florida Free-tailed Bat (*Tadarida brasiliensis* ssp. *cynocephala*) with *M. marinum* was demonstrated. Several bats developed footpad lesions (site of infection) similar to those in mice, but none developed significant visceral infections within the period they were kept alive, usually 5–14 days after the infection [33]. A later investigation involved Straw-colored Fruit Bats (*Eidolon helvum*) artificially infected with *M. ulcerans*, and re-isolation of this mycobacterial species from web muscles 4 weeks after infection [34]. In 1972, bats were considered as a possible reservoir of *M. ulcerans* infection in Australia [35], although this was not confirmed [36].

Microbiome analyses revealed the abundance of phylum Actinobacteria (including family Mycobacteriaceae) in subtropical and tropical bats’ droppings (pellets) examined 2 h after defecation or in bat guano (bat feces older than 2 h). Isolates from this phylum made up 64.1% [37], 30.0% [38], and 22.6% [39] of the microbiome from three studies in India, and 34.9% in a study from the Philippines [40]. In contrast, in Finland, phylum Actinobacteria represented less than 0.01% of the microbiome in bat guano [16].

Actinobacteria were observed in the gastrointestinal tract of bats in China. In this study, Actinobacteria were present in only 2.4% of stomach samples but were present in 15.7% of intestinal samples [41]. In contrast, Selvin et al. [42] detected Actinobacteria in 32.5% of guano samples along with Proteobacteria and Firmicutes in 70.9% and 24.9% of guano samples, respectively, however very low numbers of Actinobacteria were present in the gut samples of bat (*Rhinolophus monoceros*) in another study in India.

In a previous study, we cultured *M. fortuitum* and *M. peregrinum* from guano samples from one cave in a Moravian Karst [43]. However, genus *Mycobacterium* is immensely large and comprises more than 195 currently known validated sp. and ssp. From a clinical point of view, only eight species including *M. tuberculosis* and *M. bovis* are serious human and animal pathogens (List of Prokaryotic Names with Standing in Nomenclature; LPSN) [44] also known as tuberculous mycobacteria [36]. These eight species (*M. africanum*, *M. bovis*, *M. caprae*, *M. leprae*, *M. pinnipedii*, *M. tuberculosis*, *M. microti,* and *M. ulcerans*) belong to the risk group 3 of biological agents that can cause severe human disease and are at risk of spreading into the community; although effective prophylaxis or treatment is usually available (Directive 2000/54/EC and LPSN) [44,45]. The remaining 187 sp. and ssp. (95.9%) belong to a large group of so-called nontuberculous or environmental mycobacteria [36]. According to the clinical relevance to humans and animals, 99 (50.8%) of NTM sp. and ssp. are in the risk group 1 of biological agents that are unlikely to cause human diseases. A total of 86 (44.1%) of NTM sp. and ssp. are in risk group 2 of biological agents that can cause human disease but that are unlikely to be spread to the community and there are usually effective prophylaxis or treatment available. Only *M. yongonense* has yet to be evaluated and designated a risk group classification as of July 2021 (Directive 2000/54/EC and LPSN) [44,45].

Physico-chemical parameters (pH, organic carbon, etc.) were studied in bat guano collected from caves in a Slovak Karst (*Domica* Cave) in relation to diversity of Archaeal community [46]. In Italy, the geochemistry (pH, electrical conductivity etc.) of guano was studied in relation to microbial diversity of cave waters in the gypsum karst aquifer [47]. There is the question: which chemical and physical conditions in bats’ guano can protect mycobacteria survival or even stimulate the mycobacteria growth?

Underground and sheltered spaces including natural caves and man-made constructions in temperate zones can provide stable shelters that are regularly and repeatedly used by various bat species. Bats roost in these areas seasonally and consequently they play an important role in the dispersal of microorganisms. According to published data and our results, bat guano could be an important source of bacteria; we are specifically looking at the prevalence of NTM in bat guano. In this study, the aim was the detection, quantification, and species identification of mycobacteria present in bat guano collected from caves as well as attics and roof structures of buildings where bats roost in different European countries.

## 2. Materials and Methods

A total of 281 guano samples (bat droppings older than 2 h) from eight European countries (Bulgaria, Czech Republic, France, Hungary, Italy, Romania, Slovakia, and Slovenia) were collected and examined; 181 from caves located in all eight countries and 100 from attics and roof structures located only in the Czech Republic (Table 1 and Figure 1). Most of the guano samples collected from caves came from hibernating Lesser Horseshoe Bats (*Rhinolophus hipposideros*) or Greater Mouse-eared Bats (*Myotis myotis*), the two species most commonly found in hibernating communities [48]. Whilst these samples were collected under colonies of these bat species, the presence of guano from other bat species could not be excluded (unpublished observation). Guano collected from attics roof structures originated from summer colonies of the female Greater Mouse-eared Bats (*Myotis myotis*) with offspring in different buildings such as post offices, elementary schools, chateaus, churches, old town halls, private and residential houses, and hotels (Table 1).

### 2.1. Sample Collection

Each guano sample was separately taken by sterile tongue depressor and put into a 200 mL sterile plastic bag. After collection, the samples in plastic bags were transported to the laboratory at +6 °C where they were divided in to three parts for bacteriological analysis. Approximately 10 g of guano was transferred into a sterile disposable plastic container (30 mL) and was kept at +6 °C for up to 1 week before bacteriological analysis. Approximately 0.1 g of guano was put in a microfuge tube before DNA isolation, which was either completed immediately or stored at −20 °C until used. The remainder of the guano sample was dried for geochemical analysis [43].

### 2.2. Bacteriological Methods for Mycobacteria Detection and Identification

All 281 samples were examined for the presence of mycobacteria by direct microscopy after Ziehl–Neelsen (ZN) staining and culture examination. From 149 randomly selected guano samples, mycobacterial DNA was isolated for direct identification and quantification by qPCR (Table 2).

#### 2.2.1. ZN Staining

Before culture examination, guano samples were stained according to ZN and examined by light microscopy for the presence of acid-fast bacteria (AFB). At least 200 fields of view were examined in each guano sample [49]. The amount of AFB was evaluated as follows: negative (No AFB), + (sporadic presence of AFB), ++ (a small but significant amount of AFB present), +++ (too numerous to count = TNTC AFB) as it is shown in Table 2.

#### 2.2.2. Culture Examination

Guano samples were decontaminated using 4% NaOH and the 1% surfactant tetradecyltrimethylammonium bromide (TDAB; Duchefa Biochemie B.V., Haarlem, The Netherlands). The pre-treatment consisted of complete homogenization of the sample (5 to 10 g) with 10 mL of distilled water and shaking at 300 oscillations/min for 30 min. After centrifugation at 500 rpm (revolutions per minute) for 10 min, the supernatant was transferred to a new container and centrifuged at 4300 rpm for 20 min. A total of 10 mL of decontamination solution was added to the sediment and the sample was homogenized by vortexing for 1 min and shaking for 10 min. After centrifugation (4300 rpm for 20 min), supernatant was poured out, the pellet was neutralized by adding 15 mL of distilled water, and the pellet was vortexed. After final centrifugation (4300 rpm for 20 min) and pouring of water, the pellet was resuspended in 2.5 mL of saline. Thus, processed samples were inoculated onto four Löwenstein-Jensen media slants culture media, which were incubated at 30 and 37 °C for 12 weeks [50]. In each sample, culture positivity was evaluated according to the number of colony forming units (CFU’s). Different types of visible single CFU in one sample were individually subcultured for further identification.

#### 2.2.3. Isolates Identification

After macroscopic and microscopic evaluation, the isolates were identified mainly by molecular biological methods. Genotype *Mycobacterium* CM, AS, and NTM/DR kits (Qiagene, Hain Lifescience, Nehren, Germany) were used for basic identification. Detailed identification of mycobacterial sp. and ssp. not included in above-mentioned commercial kits was performed by sequencing and BLAST analysis [50]. *M. avium* ssp. *avium* was further identified by the PCR method for the detection of the specific IS*901* amplicon and *M. avium* ssp. *hominissuis* for the specific IS*1245* amplicon [51].

### 2.3. Molecular Methods for Mycobacteria Quantification

In parallel with culture and microscopic examinations, detection and quantification of mycobacterial DNA was performed. Z-Path-Mycobacterium_spp detection kit for Mycobacterium (Primerdesign Ltd., Eastleigh, UK) was used and E.Z.N.A.^®^ Soil DNA Kit (Omega Bio-tek, Norcross, GA, USA) was used to isolate mycobacterial DNA according to the manufacturer’s instructions. DNA isolation was performed from a 1 mL homogenized and decontaminated guano sample (described in Section 2.2.2.). Amplification and evaluation were performed on a CFX96 real-time PCR detection system (Bio-Rad Laboratories, Hercules, CA, USA) using the following thermocycler conditions: 50 cycles at 95 °C for 10 s and at 60 °C for 60 s. Quantification of viable mycobacteria was performed by comparison with a number of standards using CFX Manager Software (Bio-Rad Laboratories, Hercules, CA, USA). Quantification is expressed in copies of a specific section of DNA per 1 mol of template [50].

### 2.4. Physico-Chemical Analyses

The samples of guano (about 100 g) were transported for physico-chemical analyses on ice in a cooler and analysed immediately after sampling. Samples were dried at 60 °C until a constant weight was achieved, ISO methods were employed for the determination of pH, Electrical Conductivity (EC), Oxidation-Reduction Potential (ORP), Total Organic Carbon (TOC), and Total Inorganic Carbon (TIC). pH and EC were measured in aqueous extracts. A WTW InoLab Multi 720 tool with a SenTix 41 electrode (WTW Ltd., Czech Republic) was used for pH measurements. For EC measurements, guano aliquots were mixed with distilled water (1:5), shaken for 30 min, and measured on a multimeter WTW Multi 3320 (BDL Czech Republic s.r.o., Turnov, Czech Republic) with a TetraCon 325 electrode (Xylem Analytics Germany GmbH, Weilheim, Germany).

TOC and TIC in the guano samples were measured using a SoliTOC Cube analyzer by temperature-dependent differentiation of Total Carbon (TC) into TOC400, ROC, and TIC900 to determine the content of TOC. The method consisted of temperature ramping from 400–900 °C at a constant heat rate (70 K/min) in oxygenated air for TOC400 and TIC900 determination with a subsequent switch to a non-oxygenated air (N_2_ alone) for accurate ROC determination. The temperature hold times applied for individual C fraction determinations were 230, 150, and 120 s, respectively, and were DIN 19539-compliant. The TOC was calculated from the sum of TOC400 and ROC. All C fractions were measured using a high-sensitivity infrared sensor. The weight of measured samples was 1 ± 0.15 g.

### 2.5. Statistical Analysis

Data analysis was performed using statistical software Statistica 13.2 (StatSoft Inc., Tulsa, OK, USA) and StatXact 12.0 (Cytel Inc., Waltham, MA, USA). *p*-values less than 0.05 were considered statistically significant. ZN microscopy and culture results could be considered as ordinal variables (ZN: negative < AFB + < AFB ++ < AFB +++; culture examination: 0 = negative < 1–10 < 11–100 < TNTC). Spearman’s correlation coefficient was used to evaluate the correlations of the results of all three methods [52]. The associations between the results of ZN microscopy, results of culture examination, or results of qPCR and the origin of the guano sample were assessed using the Mann–Whitney test; *p*-values were calculated from the exact (not from asymptotic distribution). The association between the quantitative qPCR result (positive/negative) and the origin of the sample was assessed using Fisher’s exact test. The alpha-diversity of mycobacterial sp. and ssp. isolated from guano samples collected from caves and attics was assessed using the Shannon–Wiener H index of diversity [53]. The Hutcheson test was used to compare H indexes according to the origin of the samples. The beta-diversity of mycobacterial sp. and ssp. isolated from guano samples was assessed by paired PERMANOVA (Bray–Curtis dissimilarity, 9999 permutations).

## 3. Results

### 3.1. Microscopically, Culture and qPCR Examinations

ZN microscopy was positive for mycobacteria (AFB +, AFB ++ and AFB +++) in 127 (45.2%) guano samples. A total of 74 (16.3%) samples were culture positive with varying numbers of CFU per sample ranging from a single CFU to too many CFU’s to count (TNTC; Table 2). All culturable mycobacteria belonged to risk groups 1 and 2. Mycobacteria were detected in guano samples originating from caves in all 8 countries sampled (Table 1 and Figure 1). Mycobacterial DNA was detected by qPCR in 127 (85.2%) guano samples with 51 (76.1%) and 69 (84.1%) detected in guano collected from caves and attics, respectively (Table 2).

The strength of association based on size of Spearman’s correlation coefficients between ZN microscopy and culture examination can be considered moderate to substantial. The correlation between both ZN microscopy and culture examination with qPCR detection can be considered substantial to very strong. All correlations are positive, which indicates a consistent increasing trend in all used methods for mycobacteria detection (Table 3).

The proportion of samples with negative ZN microscopy examination in bat guano collected from attics was almost double that in bat guano collected from caves (Table 4 and Figure 1). The association was confirmed by Mann–Whitney test (*p* < 0.01; U = 12641).

The proportion of culture-negative guano samples collected from attics was significantly higher than this proportion of culture-positive guano samples collected from caves (*p* < 0.01; U = 10694; Mann-Whitney test). No samples with CFU’s > 10 were detected in attic-derived guano samples (Table 5 and Figure 2).

### 3.2. Detected Mycobacterial Species and Subspecies Composition and Clinical Relevance

A total of 19 NTM sp. and ssp. belonging to three groups and four complexes were isolated from bat guano samples (Table 6). All 19 NTM sp., ssp. and complex’s were identified from guano samples from caves, in contrast, only three species (*M. arupense*, *M. terrae*, and *M. fortuitum*), one subspecies (*M. avium* ssp. *hominissuis*), and one complex (*M. terrae* complex) were found in attic derived guano. The most common isolates from both types of bat guano were identified as *M. fortuitum* (*n* = 21) and *M. avium* ssp. *hominissuis* (*n* = 11). In concordance with this, NTM biodiversity members of *M. fortuitum* group (*n* = 24), *M. terrae* complex (*n* = 22), and *M. avium* complex (*n* = 14) were observed as most abundant in both types of guano samples.

Concerning clinical relevance to humans and animals, 20 (22.5%) isolates made up of 9 sp. and ssp. belonged to risk group 1 of biological agents and 48 (53.9%) isolates from 10 sp. and ssp. belonged to risk group 2 of biological agents. None of the isolated mycobacterial species belonged to risk group 3 of biological agents, and 21 (23.6%) isolates were not precisely identified (Table 6).

The alpha-diversity of NTM was statistically significantly higher for guano samples collected from caves than those from attics (*p* < 0.01; t_H_ = 3.96; df = 23.45; Hutcheson *t*-test). Paired PERMANOVA testing the frequencies of individual NTM including NK (Not Known *Mycobacterium* sp.) showed statistically significantly higher sp. and ssp. abundance (beta-diversity) of NTM isolates in cave-derived bat guano samples (*p* < 0.01; F = 24.54; Table 7).

### 3.3. Mycobacteria Quantification by qPCR

The proportions of mycobacteria positive and negative bat guano samples derived from caves or attics were similar. NTM were detected in 58 (86.6%) bat guano samples collected from caves and in 69 (84.2%) bat guano samples collected from attics (Table 8). An association between positive qPCR result and the origin of the sample could not be demonstrated (*p* > 0.05, Fisher’s exact test).

The geometric mean of cp/mL DNA NTM detected by qPCRs in cave-derived bat guano samples was approximately 7-fold higher than the geometric mean of cp/mL DNA NTMs detected in guano samples derived from attics. The median qPCR-positive bat guano samples collected from caves was about 16-fold higher than the median positive of bat guano samples from attics; 95% confidence intervals of geometric means or medians did not overlap. The median qPCR values were statistically significantly higher in cave-derived bat guano samples than those collected from attics (*p* < 0.01; Mann–Whitney test; U = 1276). Thus, not only higher alpha- and beta-biodiversity was demonstrated in bat guano samples collected from caves, but also a higher proportion of positive microscopic findings after staining according to ZN and culture examination. In the case of qPCR detection and quantification, the proportions of guano samples positive for NTM detection did not differ statistically significantly. The levels of mycobacterial DNA (and by inference the amount of viable mycobacteria) detected in cave-derived bat guano samples were an order of magnitude higher than those collected from attics (Table 9 and Figure 3).

### 3.4. Physico-Chemical Analyses of Bat Guano Samples

Statistically highly significant differences (*p* < 0.01; Mann–Whitney test) between cave and attic-derived bat guano samples were demonstrated for the EC, TC, TOC, and TIC parameters. In contrast, no statistically significant difference (*p* > 0.05; Mann–Whitney test) was found for pH and ORP (Table 10 and Figure 4).

## 4. Discussion

NTM were cultured from 32% of bat guano samples collected from different karst systems in Bulgaria, the Czech Republic, France, Hungary, Italy, Romania, Slovakia, and Slovenia (Table 1). These results correlate with finding from our previous study [43] and bat guano studies conducted in other countries including India [37,38,39], the Philippines [40], and Finland [16].

In this study, guano was mainly collected from caves where Horseshoe Bats (*Rhinolophus* spp. specifically, *Rhinolophus hipposideros* in the Czech Republic and Slovakia, and *Rhinolophus ferrumequinum* and middle sized *Rhinolophus* species in other studied European countries) were hibernating (unpublished observations). Colonies of Greater Mouse-eared Bat (*Myotis myotis*) were less commonly observed hibernating in caves in Czech localities; despite this, a number of guano samples from this species were collected from caves from this species. To complete the picture regarding the prevalence of mycobacteria in bat guano from different roosting environments, attics and other roof structures with roosting female colonies were visited. Specifically, colonies of female Greater Mouse-eared Bats (*Myotis myotis*) were found in attics roosting and giving birth during the summer season. Guano from such colonies was collected (Table 1). Dust in these spaces could also be an important reservoir of mycobacteria as well as a potential hazard for the dispersal of pollutants including metals [54,55]. Our results demonstrated culturable NTM in 16.0% of attic-derived guano samples (Table 1).

The detection of clinically relevant (risk group 2) species (i.e., *M. arupense*, *M. avium* ssp. *hominissuis*, and *M. fortuitum*) can pose a health risk for immuno-compromised people in contact with this bat guano found in attics (Table 6) [56]. In caves, bat guano was contaminated by NTM from risk group 2 such as *M. abscessus* ssp. *bolletii, M. avium* ssp. *avium, M. chelonae, M. mucogenicum, M. nonchromogenicum, M. peregrinum,* and *M. septicum* (Table 6). This data demonstrates that cave systems visited by bats could act as reservoirs of potentially pathogenic mycobacteria.

Concerning guano structure, the diet of the Lesser Horseshoe Bat consists primarily (over 50%) on Lepidoptera and Dipteran insects [57,58]. Similarly, the diet of the larger species of *Rhinolophus* genus [59] consists mostly of moths (Lepidoptera), although a ground gleaning strategy has also been observed [60]. However, the Greater Mouse-eared Bat (*Myotis myotis*) is almost exclusively a surface gleaner, foraging for flightless insects, esp. carabid beetles, crickets, and spiders [61,62]. Insectivorous bats digest chitin in the stomach using acidic mammalian chitinase [63]. However, not all chitinous parts of the body of invertebrates are digested; debris of legs, wings, feelers, rafters of beetles were observed in guano by the naked eye (unpublished observations). Currently, bat guano is considered as a new and attractive source of chitin and chitosan for cosmetics, pharmacy, and medicine etc. [64].

Zingue and colleagues demonstrated that *M. ulcerans* reservoirs are connected with chitin sources around paddy fields and swampy areas [65]. Chitin may be an important nutrient source for this mycobacterial species. The addition of chitin to media purportedly improved culture examination of *M. ulcerans* by enhancing mycobacterial growth and contributing to the maintenance of an acidic pH [66,67]. Species of Genus *Mycobacterium* play an important role in decomposition of organic waste in the environment; along with fungi, they are able to decompose materials such as polysaccharides (chitin, cellulose, etc.), lignin, and others. A lot of known enzymes involved in chitin and cellulose degradation are classified in the members of the *Mycobacterium* genus [36,68]. This information about the metabolic abilities of members of family Mycobacteriaceae allows interpretation of the abundant NTM species composition in guano isolated from cave and attic-derived guano (Table 6). Such NTM species spectrum is new information predicting an important role of NTM in bat guano degradation.

Currently, it is not clear as to how bats are exposed to mycobacteria. The most likely route of exposure is from the ingestion of non-vertebrates carrying mycobacteria. In previous studies, we demonstrated NTM could be cultured from the different non-vertebrates. *M. avium* ssp. *avium, M. avium* ssp. *hominissuis, M. avium* ssp. *paratuberculosis*, *M. avium* complex, *M. chelonae, M. fortuitum, M. phlei, M. scrofulaceum, and M.* sp. were isolated from adults of Order Diptera (families with percentages of culture positivity: Calliphoridae = 3.9%, Drosophylidae = 12.5%, Muscidae = 6.4%, Scatophagidae = 8.6%, and Syrphidae = 2.0%). *M. avium* ssp. *hominissuis, M. avium* ssp. *paratuberculosis*, *M. scrofulaceum, M. gastri, M. terrae, M. abscessus,* and *M.* sp. were isolated from Order Opisthopora (specifically individuals from the family Lumbricidae; culture positivity = 8.2%). From adult spiders and spiders’ webs of Order Araneae (culture positivity = 10.1%,), *M. avium* ssp. *avium, M. avium* ssp. *hominissuis, M. avium* ssp. *paratuberculosis*, *M. avium* complex, *M. fortuitum**, M. chelonae, M. triviale, M. xenopi,* and *M.* sp. were isolated. *M. avium* ssp. *avium, M. avium* ssp. *hominissuis, and M. avium* ssp. *paratuberculosis* were back isolated from developing stages, and imagoes of Order Coleoptera, family Tenebrionidae were fed on experimentally infected bran with the same mycobacterial ssp. [69,70,71,72,73,74,75,76,77,78,79,80].

Transmission of mycobacteria via their gastrointestinal tract was confirmed by Actinobacteria detection in bats in China with 2.4% positivity in stomach samples and 15.7% positivity in intestinal samples [41]. Why are mycobacteria an important part of the microbiome in guano? Family Mycobacteriaceae belong to phylum Actinobacteria, which is widely spread in soil, surface water sediments, and water biofilms [36]. Members of genus *Mycobacterium* (esp. *M. avium* complex) are able to survive and propagate in such oligotrophic environments [81].

The clinical relevance with respect to classification into a risk group of biological agents for humans and animals was evaluated in identified species and ssp. according to the EC law (Directive 2000/54/EC and LPSN) [44,45]. The proportion of species and ssp. that belong to risk group 1 (environmental saprophytes) and risk group 2 (potentially pathogens) was almost equivalent (Table 6). No isolate belonged to risk group 3 (obligatory pathogens including *M. tuberculosis*, *M. bovis*, *M. africanum*, *M. caprae,* and other members of *M. tuberculosis* complex.) This favorable epidemiological finding could possibly be attributed to successful control programs against human and bovine tuberculosis in all studied countries [82].

The differences in the spectrum of mycobacteria detected in bat guano from the two different environments (caves and attics; Table 6) can be explained by a number of key biological and physical factors. Specifically, the physical conditions of the environment, the activities of the bats at the time which affects their commensal and other accompanying intestinal microflora such as winter hibernation vs. summer activities, and physiology of the detected mycobacterial sp. and ssp.

In particular, temperature and humidity are essential for the metabolic activity of mycobacteria. The air temperature in the caves of the Czech Republic fluctuated between 6.2–10.6 °C. The temperature of the cave bedrock (underlying rocks and sediments) was slightly lower and ranged between 5.7–10.5 °C. Only in one cave (Zbrasov Aragonite Caves) did the temperature reach as high as 14.3 °C. [83]. Some mycobacteria (i.e., members of *M. avium-intracellulare-scrofulaceum* complex) can begin to grow at temperatures as low as 15.5 °C [84]. At these low temperatures, mycobacteria may not grow but they can remain viable for a long time (months and even years) in living stage [36,85]. Environmental saprophytic mycobacteria can propagate when the air temperature rises above 18 °C [85].

Humidity in these caves in the Czech Republic ranged between 94.6–99.5% [83]. In such high humid conditions, mycobacteria can sufficiently survive and multiply [36,85].

Conversely, the collection of attic-derived guano occurrs during summer, when the females Greater Mouse-eared Bats (*Myotis myotis*) give birth to young. At this time, temperatures in European attics can reach temperatures as high as 40 °C with relatively low humidity, typically between 25–60% [86]. These environmental conditions limit the metabolism and in some cases they can reduce the concentrations of living mycobacteria as well as degrade remnant mycobacterial DNA [36,85].

Another fundamental difference is the behaviour and activities of bats at the two types of locations. In our study, guano was collected from caves mainly during winter when bats were predominantly hibernating, whereas collection of guano from attics occurred during summer when the bats were fully active, including giving birth and taking care of young, resulting in a higher frequency of feeding and subsequent feces excretion. Furthermore, guano collected from summer colonies of the Greater Mouse-eared Bat (*Myotis myotis*) was subjected to more efficient decomposition compared to cave-derived guano. This was evident by statistically significantly higher values of conductivity and the presence of carbon in various forms (TC, TOC, TIC) in guano derived from attics compared to that from caves (Table 10 and Figure 4).

The degree of metabolism of active bats compared to those hibernating would affect the intestinal microflora, which accelerates the decomposition processes of the intestinal contents. Higher conductivity and the presence of a higher content of OC indicate the intense activity of microbes decomposing guano derived from the more active bats residing in attics. Other fast-growing intestinal microflora typically present include *Clostridium* sp., *Bacillus* sp., *Proteus* sp., *Paenibacillus* sp., *Corynebacterium* sp., and *Flavobacterium* sp. as well as other coliform Gram-negative rods that are able to degrade culture media in vitro in competition with mycobacteria [50].

In contrast, the cold environment of the caves results in suppression of the metabolic activity of the bat microflora; this was evident by a statistically significant lower EC, and lower concentration of TC and TOC, but a higher concentration of TIC (Table 10 and Figure 4). In the cold cave environment, guano remains intact for a long time [83] and consequently mycobacteria are likely able to remain viable within the guano for a substantial length of time (Table 1 and Table 2). Under these conditions, significant active multiplication of mycobacteria would not be expected; in fact, it would be expected that long-term survival would have associated with a reduction in metabolism bordering on dormancy [36]. This assumption explains the higher uptake of mycobacterial DNA (Table 9 and Figure 3) and more frequent culture isolation of mycobacteria (Table 7). In contrast, guano derived from attics is subject to faster decomposition, which would result in viable mycobacterial cells being gradually outcompeted by fast growing bacteria and subsequently eliminated, as well as the rapid degradation of residual mycobacterial DNA (Table 7 and Table 9 and Figure 3).

No differences were found between ORP and pH values between cave or attic-derived guano samples (Table 10 and Figure 4). This was probably due to the very similar molecular composition of the organic and inorganic components in both guano samples groups. These parameters may have a partial effect on the presence of living and/or unviable mycobacteria; to confirm this hypothesis, further investigation will be required not only in guano but also in fresh bat fecal samples. The ability of mycobacteria to tolerate variable pH is species specific [50]. However, the optimal pH range for most species is 5.4–7.4 [87].

With regard to the sp. and ssp. of mycobacteria isolated in this study, the pH values in both types of guano samples probably do not represent a limiting factor for their survival and possible multiplication (Table 6) [50]. ORP values in both types of guano indicated a slightly anoxic to oxidizing environment (Table 10 and Figure 4). These conditions are also unlikely to prevent the survival of mycobacteria. Again, further research is required to determine whether these physicochemical conditions effect the multiplication of other species of microorganisms that complicate culture detection of mycobacteria; frequent contamination of culture media during incubation complicates research in this field.

The specialization of various species of mycobacteria in dynamically changing environmental conditions has been demonstrated and is related to their genetic plasticity [88]. Some species of mycobacteria are able to respond more effectively to changes in the environment and adapt relatively quickly to new ecological conditions. Of the species of mycobacteria isolated in this study (Table 6), the most adaptable species is the potentially pathogenic *M. avium* ssp. *hominissuis*. This is a cosmopolitan ssp. that is able to colonize both various environmental niches and humans [36,85,89,90].

Environmentally saprophytic mycobacterial species detected in guano from both sources were *M terrae* and *M. arupense* (belonging to *M. terrae* complex), and *M. fortuitum* (Table 6). These mycobacterial species were apparently able to withstand competition with other microorganisms and, at a certain stage of guano decomposition, to temporarily predominate and/or coexist with the rest of the microflora. Generally, mycobacteria are slow growing, which is attributed to the construction of a complex cell wall and inefficient metabolism [36,85], however, it is thought they play an important role as “pioneers” in the settlement of new substrates, making unhospitable conditions more amenable for growth of other bacterial species; the cave environment is such an environment, with its extremely low levels of organic matter [83].

The amount of mycobacteria captured and their species spectrum in guano may not be final. Due to the physiological properties of mycobacteria (their slow metabolism and long generation time), the processing of highly microbially loaded samples is very difficult. The differences in the detection of mycobacteria by the methods we use are obvious and statistically substantiated (Table 2). Microscopic examination has variable but in many cases low sensitivity [91].

In addition to mycobacterial cells, a number of G + microbial flora and artifacts are present in environmental matrices, which can also be stained red after using the ZN method and can complicate the objective evaluation of the presence of mycobacteria; so-called false microscopic positivity. The qPCR method used has acceptable specificity and higher sensitivity [50]. However, as with microscopic examination, it is not possible to assess the viability of captured mycobacteria using the standard qPCR method. Dead mycobacterial cells or only residues of their genetic material may be present in the samples. Of course, the specificity of DNA capture in the performed method is not absolute. Non-specific sections of genetic material that are not of mycobacterial origin may be detected in the samples; so-called false qPCR negativity [92].

The culture test should have the highest sensitivity and specificity and clearly detect viable mycobacterial cells. Nevertheless, the reliability of cultivation can be limited. The sample processing process itself may affect the viability of the mycobacteria present. The aim is to achieve the lowest level of culture contamination and acceptable capture yield of the target micro-organism (mycobacteria). It can be accepted that some species of mycobacteria can be eliminated for cultivation by processing the matrix itself [93]. Further development of sufficiently sensitive and specific detection methods is therefore highly desirable.

In terms of competing with other bacteria, mycobacteria are less able to actively colonize guano and other matrices due to the physiological properties described above. However, as some oligotrophic microorganisms, they are more resistant and persistent in these unfavorable conditions of karst caves [36,85]. Some positive culture-determined proven (viable) species of mycobacteria were isolated exclusively from guano collected from caves (Table 6). They are likely to be transported by bats that are infected from their food sources. Their ability to survive long term in adverse conditions (low temperature, small amounts of organic material, etc.) as well as their ability to remain dormant over a long period of time in this environment has allowed them to become important representatives of the microflora within both the natural environments as well as the digestive system of bats.

## 5. Conclusions

Nontuberculous mycobacteria (NTM) were found in both attics and caves, but the number of mycobacteria in guano from the caves was more than double that from the attic. Among the isolates, no mycobacterial species belonging to the third risk group (obligatory pathogens including *M. tuberculosis* complex members) were found, 22.5% of isolates belonged to risk group 1 (environmental saprophytes) and 53.9% isolates belonged to risk group 2 (potential pathogens). NTM were isolated from guano samples collected from caves in all eight European countries (Bulgaria, Czech Republic, France, Hungary, Italy, Romania, Slovakia, and Slovenia. The correlation of detection of mycobacteria between Ziehl–Neelsen (ZN) microscopy and culture examination and qPCR was strong. Detection of mycobacteria by ZN microscopy examination in guano collected from caves (58.6%) was more than double that detected in guano collected from attics and roof structures (21.0%; *p* < 0.01). When comparing bat guano collected from the two different sources, statistically highly significant differences (*p* < 0.01; Mann–Whitney test) were observed for the following parameters: electrical conductivity, total carbon, and total organic and total inorganic carbon. In contrast, no statistically significant difference (*p* > 0.05; Mann–Whitney test) was found when comparing the pH and oxidation-reduction potential parameters.

## Data Availability

Availability of data and materials correspondence and requests for mycobacterial isolates be addressed to the corresponding author.

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
