# Peer review of "Nontuberculous Mycobacteria Prevalence in Bats’ Guano from Caves and Attics of Buildings Studied by Culture and qPCR Examinations"

_microorganisms, 2021, doi:10.3390/microorganisms9112236_

Round 1

Reviewer 1 Report

The study by Pavlik et al. provides a comprehensive analysis of a large sample of guano collected in multiple locations in 8 European countries. The Mycobacterial population isolated from the guano samples has been deeply studied using various molecular, genetic, physico-chemical methods. Please see my minor comments:

The Introduction section could be improved by providing the reasons why this study is needed and what implications would it have for humans. You talk about that at length in the Discussion but it would be great to also mention it in the Introduction to catch the reader’s attention.

Table 2 is quite busy and hard to read

Please check the title of Table 6

I would recommend to avoid too many figures and details in the Conclusions

Author Response

Response to Reviewer 1 Comments

The study by Pavlik et al. provides a comprehensive analysis of a large sample of guano collected in multiple locations in 8 European countries. The Mycobacterial population isolated from the guano samples has been deeply studied using various molecular, genetic, physico-chemical methods. Please see my minor comments:

Point 1: The Introduction section could be improved by providing the reasons why this study is needed and what implications would it have for humans. You talk about that at length in the Discussion but it would be great to also mention it in the Introduction to catch the reader’s attention. 

Response 1: “The following text was moved from Discussion section to the end of Introductory part”: Underground and sheltered spaces including natural caves and man-made constructions in temperate zones can provide stable shelters that are regularly and repeatedly used by various bat species. Bats roost in these areas seasonally and consequently they play an important role in the dispersal of microorganisms. According to published data and our results, bat guano could be an important source of bacteria, we have specifically looked at the prevalence of NTM in bat guano.

Point 2: Table 2 is quite busy and hard to read.

Response 2: The Table 2 was reduced to two categories in section ZN staining: ZN negative (-) and ZN positive (+ to +++).

Point 3: Please check the title of Table 6

Response 3: corrected.

Point 4: I would recommend to avoid too many figures and details in the Conclusions.

Response 4: Text was reduced also according to the reviewer 2.

“Nontuberculous mycobacteria (NTM) were found in both attics and caves, but the number of mycobacteria in guano from the caves was more than double that from the attic. Among the isolates, no mycobacterial species belonging to the third risk group (obligatory pathogens including M. tuberculosis complex members) were found, 22,5% of isolates belonged to risk group 1 (environmental saprophytes) and 53.9% isolates belonged to risk group 2 (potential pathogens). NTM were isolated from guano samples collected from caves in all 8 European countries (Bulgaria, Czech Republic, France, Hungary, Italy, Romania, Slovakia, and Slovenia. The correlation of detection of mycobacteria between Ziehl-Neelsen (ZN) microscopy and culture examination and qPCR was strong. Detection of mycobacteria by ZN microscopy examination in guano collected from caves (58.6%) was more than double that detected in guano collected from attics and roof structures (21.0%; P <0.01). When comparing bat guano collected from the two different sources, statistically highly significant differences (P<0.01; Mann-Whitney test) were observed for the following parameters: electrical conductivity, total carbon, total organic and total inorganic carbon. In contrast, no statistically significant difference (P> 0.05; Mann-Whitney test) was found when comparing the pH and oxidation-reduction potential parameters.”

Reviewer 2 Report

1 There is no definition ( objective ) of the purpose of physicochemical studies of bat guano. 2. The numbering of subsections of chapter three should be ordered. 3. The conclusions largely duplicate the results. Authors should revise their conclusions based on the goals of the research. For example: 1. Nontuberculous mycobacteria were found in both attics and caves, but the number of mycobacteria in guano from the caves was more than double that from the attic.2 Among the isolates, no Mycobacteria belonging to the third risk group (Pathogens) were found

1 In the introduction to the article, the authors discuss the physicochemical examination of bat guano. There is no purpose (purpose) why these tests will be performed

2 The numbering of subsections in the "Results" section should be corrected.

  1. Conclusions should relate to the purpose of the research. Currently, the conclusions are a repetition of the results

Author Response

Response to Reviewer 2 Comments

Point 1: There is no definition (objective) of the purpose of physicochemical studies of bat guano. In the introduction to the article, the authors discuss the physicochemical examination of bat guano. There is no purpose (purpose) why these tests will be performed

Response 1: “There is the question, which chemical and physical conditions in bats’ guano can protect mycobacteria survival or even stimulate the mycobacteria growth.”:

Point 2: The numbering of subsections of chapter three should be ordered.

Response 2: corrected.

Point 3: The conclusions largely duplicate the results. Authors should revise their conclusions based on the goals of the research. For example: 1. Nontuberculous mycobacteria were found in both attics and caves, but the number of mycobacteria in guano from the caves was more than double that from the attic.2 Among the isolates, no Mycobacteria belonging to the third risk group (Pathogens) were found.

Response 3: Conclusions were completely re-written and simplified.

“Nontuberculous mycobacteria (NTM) were found in both attics and caves, but the number of mycobacteria in guano from the caves was more than double that from the attic. Among the isolates, no mycobacterial species belonging to the third risk group (obligatory pathogens including M. tuberculosis complex members) were found, 22,5% of isolates belonged to risk group 1 (environmental saprophytes) and 53.9% isolates belonged to risk group 2 (potential pathogens). NTM were isolated from guano samples collected from caves in all 8 European countries (Bulgaria, Czech Republic, France, Hungary, Italy, Romania, Slovakia, and Slovenia. The correlation of detection of mycobacteria between Ziehl-Neelsen (ZN) microscopy and culture examination and qPCR was strong. Detection of mycobacteria by ZN microscopy examination in guano collected from caves (58.6%) was more than double that detected in guano collected from attics and roof structures (21.0%; P <0.01). When comparing bat guano collected from the two different sources, statistically highly significant differences (P<0.01; Mann-Whitney test) were observed for the following parameters: electrical conductivity, total carbon, total organic and total inorganic carbon. In contrast, no statistically significant difference (P> 0.05; Mann-Whitney test) was found when comparing the pH and oxidation-reduction potential parameters.”

Reviewer 3 Report

Line 134: Table 1 is missing a title

The headings in Table 1 require reformatting so that it is much clearer which heading belongs to which column.

I wonder also about the placement of the table – which contains results. This is currently placed early in the methods section.

Line 172: AFB (acid-fast bacilli) is used much more frequently than ‘AFR’. I would suggest using ‘AFB’.

Line 182/3: rpm/min? Is ‘rpm’ revolutions per min?

Line 179 / 183: The exact composition of the decontamination solution should be provided. For example, we do not know the concentration of tetradecyltrimethylammonium bromide that is being used. The methods section must provide sufficient information to enable others to use the same protocols.

Line 185: Insufficient information is provided for centrifugation and neutralization. How did you neutralize tetradecyltrimethylammonium bromide?

Line 186: It seems that two LJ slants were used at each temperature. Were these exact duplicates? Or were different types of LJ used?

Line 249: What is the title of section 3.1?

Table 2: I find Table 2 difficult to understand. The title of table 2 refers to species diversity but the contents of table 2 shed no light on species diversity.

There are lines of numbers (at the bottom of each section in the table) without any labelling. What do these refer to?

Similarly, for the line marked with “%” (e.g. 54.8, 48.7, etc….) it is not obvious how these are derived or what they refer to. This table should be much more user friendly for readers.

The title of Table 4 is “Diagnostic sensitivity and specificity at optimal cut-off value derived for the predictive model”. However, the numbers in the table do not appear to be percentages for either sensitivity or specificity.

Table 6 has no title.

The title of table 10 refers to species diversity but the contents of table 2 shed no light on species diversity.

I find that the discussion is lacking with respect to the likely reasons for the different results achieved by different methods. For example, for samples that are PCR +  but culture negative, could this be due to killing of the NTM by the harsh decontamination strategy. There is ample evidence that can be cited to show that 4% NaOH can kill some species of NTM. For samples that were Zn + but culture-negative could this be due to species other than mycobacteria showing as AFB? Or is it further evidence of the problems caused by decontamination?

Author Response

Response to Reviewer 3 Comments

Point 1: Line 134: Table 1 is missing a title. The headings in Table 1 require reformatting so that it is much clearer which heading belongs to which column. I wonder also about the placement of the table – which contains results. This is currently placed early in the methods section.

Response 1: corrected and Table 1 and Map 1 were moved closer to the Results Chapter.

Point 2: Line 172: AFB (acid-fast bacilli) is used much more frequently than ‘AFR’. I would suggest using ‘AFB’.

Response 2: corrected.

Point 3: Line 182/3: rpm/min? Is ‘rpm’ revolutions per min?

Response 3: corrected and added explanation “500 rpm (revolutions per minute) for 10 min”.

Point 4: Subchapter 2.2.2. Culture examination

Line 179 / 183: The exact composition of the decontamination solution should be provided. For example, we do not know the concentration of tetradecyltrimethylammonium bromide that is being used. The methods section must provide sufficient information to enable others to use the same protocols.

Line 185: Insufficient information is provided for centrifugation and neutralization. How did you neutralize tetradecyltrimethylammonium bromide?

Line 186: It seems that two LJ slants were used at each temperature. Were these exact duplicates? Or were different types of LJ used?

Response 4: corrected in the text.

Point 5: Line 249: What is the title of section 3.1?

Response 5: corrected “3.1. Microscopically, culture and qPCR examinations”

Point 6: Table 2: I find Table 2 difficult to understand. The title of table 2 refers to species diversity but the contents of table 2 shed no light on species diversity.

There are lines of numbers (at the bottom of each section in the table) without any labelling. What do these refer to?

Similarly, for the line marked with “%” (e.g. 54.8, 48.7, etc….) it is not obvious how these are derived or what they refer to. This table should be much more user friendly for readers.

Response 6: corrected “Mycobacteria detection in caves’ and attics’ guano samples” and according to the reviewer 2 the Table 2 was also simplified.

Point 7: The title of Table 4 is “Diagnostic sensitivity and specificity at optimal cut-off value derived for the predictive model”. However, the numbers in the table do not appear to be percentages for either sensitivity or specificity.

Response 7: corrected “Bat guano origin and ZN microscopy results.”

Point 8: Table 6 has no title.

Response 8: corrected “Detected identified mycobacteria in relation to the complexes and groups.”

Point 9: The title of table 10 refers to species diversity but the contents of table 2 shed no light on species diversity.

Response 9: corrected “Physico-chemical parameters of bat guano samples.”

Point 10: I find that the discussion is lacking with respect to the likely reasons for the different results achieved by different methods. For example, for samples that are PCR +  but culture negative, could this be due to killing of the NTM by the harsh decontamination strategy. There is ample evidence that can be cited to show that 4% NaOH can kill some species of NTM. For samples that were Zn + but culture-negative could this be due to species other than mycobacteria showing as AFB? Or is it further evidence of the problems caused by decontamination?

Response 10: Text was added (Lines 548-571) including 3 new references (in yellow colour).

Round 2

Reviewer 2 Report

The authors corrected the text in accordance with the comments of the reviewer.

Author Response

We have added the required text (in blue colour) describing the neutralization of decontaminated sample.

“After centrifugation (4 300 rpm for 20 min) supernatant was poured out, pellet was neutralized by adding 15 ml of distilled water and pellet was vortexed. After final centrifugation (4 300 rpm for 20 min) and pouring of water, pellet was resuspended in 2.5 ml of saline. Thus processed samples were inoculated onto four Löwenstein-Jensen media slants culture media, which were incubated at 30 and 37 °C for 12 weeks [50].|”

Reviewer 3 Report

None. It would have been desirable to include more detail (as requested) on the decontamination protocol e.g. the concentration of all of the ingredients and how they were neutralized prior to culture.

Author Response

Letter to the editor

Dear Editor,

We would like to thank for both reviewers’ opinions, and suggestions. We have added the required text (in blue colour) describing the neutralization of decontaminated sample.

“After centrifugation (4 300 rpm for 20 min) supernatant was poured out, pellet was neutralized by adding 15 ml of distilled water and pellet was vortexed. After final centrifugation (4 300 rpm for 20 min) and pouring of water, pellet was resuspended in 2.5 ml of saline. Thus processed samples were inoculated onto four Löwenstein-Jensen media slants culture media, which were incubated at 30 and 37 °C for 12 weeks [50].|”

We hope, that now the manuscript is finally improved.

Kind regards

Ivo Pavlik